Beneficial endophytic fungi improve the yield and quality of Salvia miltiorrhiza by performing different ecological functions

Li Xiaoyu 1 2
Lin Yali 3
Qin Yong 1 2
Han Guiqi 1 2 4
Wang Hai 4
Yan Zhuyun yanzhuyun@cdutcm.edu.cn 1 2
1 State Key Laboratory of Characteristic Chinese Medicine Resources in Southwest China, School of Pharmacy, Chengdu University of Traditional Chinese Medicine , Chengdu , Sichuan , China
2 School of Pharmacy, Chengdu University of Traditional Chinese Medicine , Chengdu , Sichuan , China
3 Patent Examination Cooperation Sichuan Center of the Patent Office, CNIPA , Chengdu , Sichaun , China
4 College of Medical Technology, Chengdu University of Traditional Chinese Medicine , Chengdu , Sichuan , China
U’Ren Jana
Electronic publication date: 2024 Feb 22
Publication date: 2024
Volume: 12
Electronic Location ID: e16959
Received 2023 Oct 2; Accepted 2024 Jan 25
Copyright: ©2024 Li et al.
Copyright year: 2024
Copyright holder: Li et al.
License: This is an open access article distributed under the terms of the Creative Commons Attribution License, which permits unrestricted use, distribution, reproduction and adaptation in any medium and for any purpose provided that it is properly attributed. For attribution, the original author(s), title, publication source (PeerJ) and either DOI or URL of the article must be cited.
License URL: https://creativecommons.org/licenses/by/4.0/

Keywords: Salvia miltiorrhiza, Endophytic fungi, Secondary metabolite, Medicinal plant, Plant-microbe interactions, Ecological functions, Photosynthesis

Funding: National Natural Science Foundation of China 81973416 Science and Technology Department of Sichuan Province 2021YFS0045 This research was funded by the National Natural Science Foundation of China (grant number 81973416) and the Science and Technology Department of Sichuan Province (grant number 2021YFS0045). The funders had no role in study design, data collection and analysis, decision to publish, or preparation of the manuscript.

==============================
Background

Endophytic fungi can enhance the growth and synthesis of secondary metabolites in medicinal plants. Salvia miltiorrhiza Bunge is frequently employed for treating cardiovascular and cerebrovascular ailments, with the primary bioactive components being salvianolic acid and tanshinone. However, their levels in cultivated S. miltiorrhiza are inferior to that of the wild herbs, so the production of high-quality medicinal herbs is sharply declining. Consequently, the utilization of beneficial endophytic fungi to improve the yield and quality of S. miltiorrhiza holds great significance for the cultivation of medicinal plants.

Methods

In this study, nine non-pathogenic, endophytic fungal strains were introduced into sterile S. miltiorrhiza seedlings and cultivated both in vitro and in situ (the greenhouse). The effects of these strains on the growth indices, C and N metabolism, antioxidant activity, photosynthesis, and content of bioactive ingredients in S. miltiorrhiza were then evaluated.

Results

The results showed that the different genera, species, or strains of endophytic fungi regulated the growth and metabolism of S. miltiorrhiza in unique ways. These endophytic fungi primarily exerted their growth-promoting effects by increasing the net photosynthetic rate, intercellular CO2 concentration, and the activities of sucrose synthase, sucrose phosphate synthase, nitrate reductase, and glutamine synthetase. They also enhanced the adaptability and resistance to environmental stresses by improving the synthesis of osmoregulatory compounds and the activity of antioxidant enzymes. However, their regulatory effects on the growth and development of S. miltiorrhiza were affected by environmental changes. Moreover, the strains that significantly promoted the synthesis and accumulation of phenolic acids inhibited the accumulation of tanshinones components, and vice versa. The endophytic fungal strains Penicillium meloforme DS8, Berkeleyomyces basicola DS10, and Acremonium sclerotigenum DS12 enhanced the bioaccumulation of tanshinones. Fusarium solani DS16 elevated the rosmarinic acid content and yields in S. miltiorrhiza. The strain Penicillium javanicum DS5 improved the contents of dihydrotanshinone, salvianolic acid B, and rosmarinic acid. The strains P. meloforme DS8 and B. basicola DS10 improved resistance.

Conclusion

Various endophytic fungi affected the quality and yield of S. miltiorrhiza by regulating different physiological and metabolic pathways. This study also provides a novel and effective method to maximize the effects of beneficial endophytic fungi by selecting specific strains to design microbial communities based on the different ecological functions of endophytic fungi under varying environments and for specific production goals.

Introduction

Endophytic fungi are a group of microorganisms that inhabit plants during a specific stage or throughout their life cycle without significantly harming them (Rana et al., 2020). Through recognition and selection, they establish a relatively stable and reciprocal symbiosis with the host plant (Ji et al., 2022). They perform various ecological functions, are involved in plant growth and development, and are essential for the adaptability and diversity of host plants (Yan et al., 2019). Furthermore, the composition and functions of the endophytic fungal community directly impact the growth and health of the host plants (Zhao et al., 2020). The effects of endophytic fungi on plants are diverse, with some being harmful and even causing disease or death due to environmental changes, community sensing, and other factors (Zanne et al., 2020). Most endophytic fungi do not exhibit any prominent functional properties regarding plant growth and development and their adaptability to environmental changes (Hardoim et al., 2015). Only a few beneficial endophytic fungi have beneficial functions such as promoting growth (Khan et al., 2015), improving resistance (Verma et al., 2021), and enhancing the accumulation of secondary metabolites in host plants (Zhai et al., 2017a). Beneficial endophytic fungi (BEF) can play a crucial role in the development of symbiosis with host plants by regulating photosynthesis, promoting nutrient absorption and distribution, and enhancing growth by inducing the production of growth hormones (Sarkar et al., 2021). Additionally, BEF plays a role in stress-resistance-associated ecological functions to minimize the effects of stress-inducing conditions when the external environment alters and aids the host plants in adjusting to the new, unfavorable environments (Rodriguez & Redman, 2008). Notably, during the co-evolution of endophytic fungi with plants, they only attained the ability to produce the same secondary metabolites as their host plants (Cao et al., 2021), but can also induce or stimulate the activity of the secondary metabolite biosynthesis-related enzymes to promote their production (Chen et al., 2022). These bioactive ingredients are usually advantageous to improving the quality formation of medicinal plants(Ming et al., 2013). Consequently, BEF can be a useful method to promote plant growth and development and increase the contents of bioactive ingredients and demonstrate multiple beneficial functions. These make their use in the cultivation of crops and medicinal plants promising, thereby attracting the attention of researchers from various disciplines, including botany and ecology, in recent years.

The dried roots and rhizomes of the perennial herb Salvia miltiorrhiza Bunge., known as danshen in China, have the effects of invigorating blood stasis, relieving pain, clearing the heart, cooling the blood, and eliminating carbuncles (MEIm et al., 2019). It is an essential herbal medicine commonly used in treating cardiovascular and cerebrovascular diseases in China for more than 2,000 years (Li, Xu & Liu, 2018b). The pharmacologically active components of Salvia belong to two main categories: fat-soluble tanshinones and water-soluble phenolic acids (Su et al., 2015). The fast-paced modern lifestyle of humans frequently leads to cardiovascular and cerebrovascular diseases, which seriously affect life and health. Additionally, the demand for S. miltiorrhiza and its bioactive ingredients has enhanced due to the rapid development of the pharmaceutical industry. All these pressures have led to an increased demand for Salvia-based herbal medicines. Unfortunately, artificial cultivation fulfills most of the market’s demand for S. miltiorrhiza due to a scarcity of wild resources (Cui et al., 2021). In China, S. miltiorrhiza is cultivated in an area of 40,000 hm2 with a production of 20,000 tons (Jiang, Tian & Li, 2018). However, the levels of bioactive ingredients in the cultivated S. miltiorrhiza plants were lower than those found in the wild due to technological limitations and environmental influences (Liang et al., 2021). Using efficient methods to obtain high-yield and good-quality medicinal plants is a scientific problem that has to be addressed urgently. A close relationship between the S. miltiorrhiza-associated endophytic fungi and the amounts of bioactive ingredients in roots has been demonstrated (Ding et al., 2015; Sun et al., 2014). For instance, Cladosporium sp. SM58 (Zhou, Tang & Guo, 2018a), and Alternaria sp. A13 (Zhou, Tang & Guo, 2018b) facilitated the growth of S. miltiorrhiza. Specifically, Trichoderma atroviride D16 (Ming et al., 2013), Mucor circinelloides DF20 (Chen et al., 2021), and Cladosporium tenuissimum DF11 (Chen et al., 2022) elevated the tanshinone content in S. miltiorrhiza, while Chaetomium globosum D38 (Zhai et al., 2017b) and M. fragilis DF20 (Xu et al., 2021) enhanced those of salvianolic acids. Although endophytic fungi have demonstrated a good applicability value in improving the quality and yield of S. miltiorrhiza, only a limited number of beneficial fungi have been identified. Most strains enhance only the yield or the tanshinone or phenolic acid contents of S. miltiorrhiza, but not all three together. Therefore, screening BEF strains to improve the yield and quality of S. miltiorrhiza herbs and elucidating the mechanisms by which exert their probiotic effects is of great value in crop production.

Materials & Methods

Test materials and pretreatment

(1) Plant sample

S. miltiorrhiza was obtained through cultivation of sterile test-tube seedlings in vitro. The explants, sourced from young leaves of Zhongjiang Daye S. miltiorrhiza grown in the Medicinal Botanical Garden of the Chengdu University of Traditional Chinese Medicine in Sichuan, Wenjiang, was identified as S. miltiorrhiza Bge. by professor Zhuyun Yan (Chengdu University of Traditional Chinese Medicine, College of Pharmacy). The S. miltiorrhiza sample was kept at State Key Laboratory of Characteristic Chinese Medicine Resources in Southwest China, Chengdu, China, and labeled SMDY07. The explants were sterilized and placed in callus medium consisting of MS + 2.0 mg  L−16-BA + 1.0 mg  L−1 NAA + 3% sucrose + 0.6% agar. The culture was maintained in the dark at a temperature of 25 °C. Once the tissue had dedifferentiated into a callus, it was transferred to the bud medium consisting of MS + 2.0 mg  L−16-BA + 0.2 mg  L−1 NAA + 3% sucrose + 0.6% agar with light intensity set at 3000 lx and light time set at 14 h daily. After the buds had grown tall, S. miltiorrhiza seedlings with 2–4 tender leaves were taken and inserted into the rooting medium consisting of 1/2 MS + 0.2 mg  L−1 NAA + 0.5 mg  L−1 IBA + 1.5% sucrose + 0.6% agar. When 3–5 roots have grown, it can be used as a test tube seedling.

(2) Soil sample

The potting soil was sandy paddy soil taken from the Medicine Plant Garden of Chengdu University of Traditional Chinese Medicine (Sichuan, Wenjiang, China). The content of organic matter in the soil was 25.58 g  kg−1, the content of available phosphorus was 98.42 mg  kg−1, the content of available potassium was 98.75 mg  kg−1, the content of alkaline hydrolysis nitrogen was 94.41 mg  kg−1, and the pH was 7.3. The soil was passed through a No. 4 sieve with a pore size of 0.23 mm, sterilized by autoclave at 121 °C for 3 h, and passed sterility tests before use. Perlite was soaked in 75% ethanol for 12 h before air-drying. The sterile soil and perlite were mixed evenly at a 1:1 volume ratio and then put into pots (20 cm in diameter and 30 cm in height), each pot weighing 6 kg.

(3) Fungal materials

The endophytic fungi in this experiment were non-pathogenic endophytic fungi isolated, verified, and preserved from the roots of S. miltiorrhiza in the early stages of the research group. ITS sequences of endophytic fungi were extracted by referring to the method of Martin, K.J (Martin & Rygiewicz, 2005). Then ITS sequences were matched with homologous sequences by BLAST searches, using the Kimura 2-parameter model to calculate the distances and establish an evolutionary relationship by neighbor-joining cluster analysis. Finally, the ITS sequences of nine strains were uploaded to NCBI GenBank to obtain accession numbers. See Table 1 for information on the strains tested. The preserved strains were activated on a PDA medium before being used. After 2-3 days of incubation at 37 °C, the strains’ growth was checked to ensure no contamination from other microbes.

Table 1 Information about the test strains.

Strain number	Strain information	GenBank accession number	
DS5	Penicillium javanicum	OP547504	
DS7	Acremonium sclerotigenum	OP547505	
DS8	Penicillium meloforme	OP547506	
DS10	Berkeleyomyces basicola	OP547507	
DS12	Acremonium sclerotigenum	OP547508	
DS13	Fusarium solani	OP547509	
DS16	Fusarium solani	OP547510	
DS17	Purpureocillium lilacinum	OP547511	
DS18	Clonostachys rosea	OP547512	

Experimental design

In vitro culture: Test tube-grown seedlings with three to five roots were placed on supports and transferred to sterilized culture flasks (6 × 10 cm) containing 100 mL of 1/2 MS liquid medium. Following three days of culture, the roots of S. miltiorrhiza were carefully pierced using a sterile needle and inoculated with a piece of fungal cake, six mm in diameter. The control group plants were treated similarly but inoculated with a bit of PDA medium, six mm diameter. Each treatment was conducted with each endophytic fungus in triplicates. After the seedlings were cultured for 30 days, the culture medium and mycelium attached to the roots were cleaned, and the net weight gain, increase in height, and number of new leaves of the seedlings were measured. The roots were then cleaned five times with sterile water, cut into five mm pieces, and dissociated in a 100 g L−1 KOH at 100 °C in a water bath for one hour. The successful colonization was verified by staining with 0.5 g L−1 acidic magenta followed by decolorization with lactic acid glycerol and then observed under a microscope (Phillips, 1970). The colonization of the nine endophytic fungal strains is shown in Fig. S1.

In situ (the greenhouse): In line with the sterile culture methods, each endophytic fungus was cocultured with S. miltiorrhiza for seven days, which confirmed the successful colonization. The control group plants were cocultured with the non-inoculated PDA medium for seven days. After three days of acclimatization, the seedlings were transplanted into pots filled with sterilized soil, sufficiently irrigated with sterile water, and regularly supplied with liquid MS solution and sterile water. The experiments used three pots for each group and two seedlings in each pot. The seedlings were cultivated for 36 weeks in an outdoor greenhouse under natural light and temperature. After harvest, the plants were removed from the pots, rinsed with tap water, and the excess water was dried using filter papers. One plant from each pot was dried at 105 °C for 12 h, and after drying to a stable weight, the dry weight and biomass of the roots and rhizomes (excluding fibrous roots) were measured. The remaining root tissue was stored at −80 °C and then used to determine the levels of the bioactive components.

Indole-3-acetic acid (IAA) content assay

Once the preserved strains were activated, individual colonies were carefully selected and inoculated onto PDA plates. As soon as the colonies had grown to a diameter of 0.5 cm, two mm fragments of the fungi were transferred into shaking flasks that were pre-filled with 50 mL of King’s culture solution. To determine the IAA content in the culture solution, the Salkowski colorimetric method was employed, and each strain was replicated three times and cultivated for a period of 12 days at 28 °C with 120 rpm shaking, in accordance with the methodology outlined by Gang et al. (2019).

Photosynthesis parameter measurement

The content of chlorophyll in leaves was determined by spectrophotometry (Moran & Porath, 1980). The net photosynthetic rate Pn (µmolCO2 m−2 s−1), stomatal conductance Gs (mmolH2O m−2 s−1), intercellular CO2 concentration Ci (µmolCO2 mol−1), and transpiration rate Tr (mmolH2O) were measured by the LI-6400X portable photosynthetic meter (LI-COR, USA).

Determination of enzyme activities and products related to C and N metabolism

The activities of sucrose synthase (SS) and sucrose phosphate synthase (SPS) were assessed according to the method described by Guy, Huber & Huber (1992) and Wang et al. (1993). Briefly, 0.5 g of fresh leaves were taken and placed in a pre-cooled mortar. The leaves were then ground in 100 mmol L−1 Hepes-NaOH (pH 7.5) buffer under ice bath conditions. The resulting extract was centrifuged at 4 °C and 10,000 rpm for 10 min, yielding the SS and SPS enzyme extracts. The enzyme extract was then added to a 100 mmol L−1 Hepes-NaOH (pH 7.5) buffer containing 50 mmol L−1 MgCl2, 100 mmol L−1 uridine diphosphate glucose (UDPG), and 100 mmol L−1 fructose, and the mixture was reacted in a water bath at 30 °C for 30 min. The reaction was terminated by adding two mmol L−1 NaOH, 0.1% resorcinol, and concentrated hydrochloric acid. The mixture was then held in a water bath at 30 °C for 10 min, and the absorbance at 480 nm was measured after cooling. SPS enzyme activity was determined by replacing fructose with fructose 6-phosphate in the above-mentioned method, followed by heating in a water bath at 80 °C for 10 min.

The soluble sugar content was determined using the method described by Zhang et al. (2019). Fresh leaves (0.1−0.3 g) were placed in a 20-mL graduated test tube, and 5-10 mL of distilled water were added. The tube was then sealed and placed in a boiling water bath for 5 min, 2-3 times in total. After cooling, the filtrate was collected in a 25-mL volumetric flask, and the volume was adjusted. Then, 0.5 mL of the extract, 1.5 mL of water, five mL of ethyl anthrone acetate, and five mL of concentrated sulfuric acid were added to a 20-mL stoppered graduated test tube. The contents were fully shaken, followed by warm incubation for 1 min in boiling water and natural cooling. The absorbance at 620 nm was then measured, using a blank as a control.

The reducing sugar content was measured using the Lindsay (1973) method. Specifically, 0.5 g of S. miltiorrhiza leaves were placed in a test tube, followed by 5-6 mL of 80% ethanol. The test tube was then placed in a water bath at 80 °C for 30 min, while continuously stirring. After centrifugation at 3,500 rpm for 15 min, the supernatant was collected, and the extraction process was repeated. The supernatants were mixed and diluted to 25 mL with distilled water, and the extract was evaporated in a boiling water bath. The residue was then dissolved with stirring in 10-20 mL of distilled water, followed by centrifugation. The test tubes were filled with two mL of supernatant and two mL of DNS, and the mixture was heated in a boiling water bath for 5 min. The test tubes were immediately cooled to room temperature in cold water, diluted with distilled water to 25 mL, shaken, and the absorbance at 540 nm was measured with a blank as the control.

The soluble protein content and nitrate reductase (NR), glutamine synthase (GS), and glutamate synthase (GOGAT) activities were determined according to the approach of Zhu et al. (2021). Fresh leaves weighing 0.5 g were first placed in a pre-cooled mortar and then mixed with quartz sand and four mL of extraction solution before being ground in an ice bath. The resulting mixture was subjected to centrifugation at 4 °C and 4,000 rpm for 15 min to obtain the NR crude enzyme extract, which was followed by the addition of 1.2 mL of KNO3 and 0.4 mL of NADH to 0.4 mL of crude enzyme solution. While keeping the solution warm at 25 °C for 30 min, a phosphate buffer solution with a pH of 7.5 was used as a control in place of NADH. The reaction was halted with one mL of sulfonamide solution, and one mL of naphthyl vinylamine was added before centrifuging at 4,000 rpm for 15 min. The absorbance at 540 nm of the resultant supernatant was measured.

Similarly, GS crude enzyme extract was obtained by mixing 0.5 g of leaves with an imidazole and hydrochloric acid buffer and subjecting the homogenate to centrifugation at 4 °C and 10,000 rpm for 15 min. 1.2 mL of crude enzyme solution was mixed with 0.6 mL of imidazole-hydrochloric acid buffer (0.25 mol L−1, pH 7.9), 0.4 mL of sodium glutamate solution (0.3 mol L−1, pH 7.0), 0.4 mL of ATP-Na solution (30 mmol L−1, pH 7.0), and 0.2 mL of MgSO4 (0.5 mol L−1). After the mixed reaction solution had been in a water bath at 25 °C for 5 min, the reaction started with 0.2 mL of hydroxylamine reagent and ended with 0.8 mL of acidic FeCl3 reagent after 15 min. The mixture was centrifuged at 4,000 rpm for 15 min, and the supernatant’s absorbance at 540 nm was measured.

Moreover, the GOGAT crude enzyme extract was obtained by mixing 0.5 g of S. miltiorrhiza leaves with two mL of imidazole-hydrochloric acid buffer (pH 7.2), followed by grinding in an ice bath, allowing it to stand for 30 min, and then centrifuging at 12,000 rpm for 20 min at 4 °C. To 0.3 mL of crude enzyme solution, 0.4 mL of 20 mmol L−1 L-glutamine, 0.5 mL of 20 mmol L−1 α-ketoglutarate, 0.1 mL of 10 mmol L−1 KCl, 0.2 mL of three mmol L−1 NADH, and 1.5 mL of 25 mmol L−1 Tris–HCl buffer (pH 7.6) were added, and the enzyme activity was measured continuously after the start of the reaction. The activity of GOGAT was expressed as the amount of NADH reduction product produced per unit reaction time.

Finally, the soluble protein content was determined using the Komas Brilliant Blue method (Bradford, 1976). To do this, fresh leaves weighing 0.5 g were ground with distilled water and centrifuged at 10,000 rpm. The supernatant was obtained as the solution to be measured to which Coomassie Bright Blue was added. After standing, the absorbance at 595 nm was measured.

Antioxidant enzyme activity assay

Referring to the method of Nahakpam & Shah (2011) to determine superoxide dismutase (SOD) activity, peroxidase (POD) activity, and catalase (CAT) activity.

An assay of SOD enzyme activity involved the addition of 0.5 g of fresh leaves to 2.5 mL of phosphate buffer (pH 7.8), followed by grinding in an ice bath. Subsequently, 2.5 mL of phosphate buffer was added, mixed, and the supernatant obtained after 4 °C and 10,000 rpm centrifugation for 15 min was the crude enzyme extract. Take 20 µL of crude enzyme solution and add it to a mixture of three mL of 2 µmol L−1 riboflavin, three mL of 75 µmol L−1 nitroblue tetrazolium (NBT), three mL of 13 mmol L−1 methionine, 0.3 mL of 0.1 mmol L−1 ethylenediaminetetraacetic acid (EDTA), and 1.5 mL of 50 mmol L−1 phosphate buffer (pH 7.8). SOD enzyme activity is defined as the unit that inhibits the reduction of NBT by 50%.

The POD enzyme activity assay encompassed the addition of 0.2 g of fresh leaves to three mL of phosphate buffer (pH 7.0) and grinding in an ice bath. The mixture was centrifuged at 4 °C at 15,000 rpm for 10 min, and the supernatant obtained was the GS crude enzyme extract. The reaction mixture involved the use of 0.15 mL of crude enzyme solution, 40 mmol L−1 of pH 7.8 phosphate buffer, 10 mmol L−1 of H2O2 solution, and 20 mmol L−1 of guaiacol. Absorbance at 470 nm was measured at 30 s and 1.5 min intervals.

The CAT enzyme activity assay comprised the mixing of 0.2 g of leaves in phosphate buffer and hydrogen peroxide solutions and grinding in an ice bath. Centrifugation was then done at 22,000 rpm for 15 min at 4 °C, and the absorbance of the supernatant was measured at 270 nm.

The assay of malondialdehyde (MDA) content was done as described by Lastochkina et al. (2017), involving the grinding of S. miltiorrhiza leaves with an extraction solution (50 mmol L−1 phosphate buffer, 1% polyvinylpyrrolidone, and 0.2 mmol L−1 ascorbic acid) in an ice bath. Centrifugation at 4 °C and 15,000 rpm for 30 min was performed to obtain the crude enzyme extract. The analysis utilized one mL of the crude enzyme solution and four mL of a trichloroacetic acid solution containing 0.5% barbituric acid. After heating at 100 °C for 30 min, the absorbance at 532 nm was measured.

Phenylalanine ammonia-lyase (PAL) enzyme activity assay (Jaafar, Ibrahim & Fakri, 2012): S. miltiorrhiza leaves were taken at 1 g and filtered by adding 10 mL of extract (borate buffer containing 50 mmol L−1 of mercaptoethanol, 0.5 g polyvinylpyrrolidone (PVP), 0.3 mL glycerol, and 0.2 g EDTA). The filtrate was centrifuged at 4 °C for 15 min at 10,000 rpm. The supernatant was the crude enzyme extract. Next, one mL of crude enzyme solution, 0.02 mol L−1 phenylalanine, and distilled water were heated at 30 °C for 30 min in a constant-temperature water bath. The reaction was then terminated by adding 6 mol L−1 trichloroacetic acid, and the absorbance was measured at 290 nm using a UV spectrophotometer.

The assay of proline (Pro) content (Bates, Waldren & Teare, 1973) entailed the addition of sulfosalicylic acid to 0.5 g of S. miltiorrhiza leaves followed by extraction in boiling water for 10 min. After subsequent filtering of the extract, the filtrate served as the measurement solution. The analyzed solution was mixed with a certain amount of glacial acetic acid and acidic ninhydrin, heated in boiling water for 30 min, and added to toluene after cooling. After standing for 10 min, centrifugation was performed, and the absorbance at 520 nm was measured.

Determination of active ingredients of S. miltiorrhiza

Referring to the method of Liu et al. (2022), the test solution was prepared, and the active ingredients in S. miltiorrhiza roots were determined by ultra-high performance liquid chromatography (UPLC). The chromatographic column was Agilent ZORBAX Eclipse Plus C18 (2.1 × 50 mm, 1.8 µm). The mobile phase was an acetonitrile (A)−0.02% phosphoric acid-water (B) system, and the chromatographic elution gradient (expressed as the proportion of the B phase) was as follows: 0∼0.5 min, 95% B; 0.5∼2 min, 95%∼87% B; 2∼6.5 min, 87%∼78% B; 6.5∼10 min, 78%∼72% B; 10∼11.5min, 72%∼40% B; 11.5∼15min, 40%∼10% B; the flow rate was 0.5 mL min−1; the detection wavelength was 280nm; and the temperature of the column was 35 °C. Then, 0.5 µL of the control solution and 1 µL of the test solution were accurately aspirated, injected into the liquid chromatograph, and determined. Qualitative identification of the components was made by comparing the retention times of the samples and standards, and quantitative calculation of the active ingredients was made by the external standard method. The UPLC chromatogram of mixed standards and samples can be seen in Fig. S2. The linear range of the standard curve, the linear regression equation, and the correlation coefficient are shown in Table 2.

Table 2 The calibration curve data of reference substance in S. miltiorrhiza root.

Active ingredients	Linear range (ng)	Regression equation	r 2	
Caffeic acid	0.168∼8.960	y = 36.946x − 2.7674	0.9999	
Rosmarinic acid	1.026∼54.720	y = 20.027x − 2.2087	0.9999	
Salvianolic acid B	91.60∼916.000	y = 6.622x − 40.736	0.9999	
Salvianolic acid A	0.420∼22.420	y = 28.296x − 4.4895	0.9997	
Dihydrotanshinone	2.019∼107.680	y = 29.376x − 19.096	0.9999	
Tanshinone I	0.252∼8.064	y = 14.921x − 1.1008	0.9996	
Cryptotanshinone	1.470∼78.400	y = 39.007x − 31.526	0.9999	
Tanshinone IIA	0.510∼27.200	y = 66.731x − 14.066	0.9999	

Statistical analysis

IBM SPSS 26 software was used for statistical data analysis, and one-way analysis of variance (ANOVA) was used for multi-sample comparison. When variances were normal and homogeneous, the Dunnett’s t test was used for multiple comparisons (P < 0.05). Graphs were drawn using GraphPad Prism 9.

Results

Beneficial endophytic fungi can promote the growth of S. miltiorrhiza

After 30 days of in vitro cocultivation, the strains DS5, DS7, DS8, DS10, DS12, and DS13 significantly promoted growth, increasing seedling weight [P < 0.05] (Fig. 1A). All strains except DS10, 17, and 18 markedly enhanced the seedling height (Fig. 1B), whereas DS12, 13, and 16 reduced the number of new leaves (Fig. 1C). After 36 weeks of cocultivation, strains DS5, 8, 10, 13, 16, 17, and 18 promoted the growth of medicinally-important parts of S. miltiorrhiza but did not reach significance. (Figs. 1E, and 2); while DS16 significantly elevated both the biomass and yield of medicinal parts. The different genera, species, or strains of endophytic fungi regulated the growth of S. miltiorrhiza in varied ways. Acremonium sclerotigenum DS7 and 12 enhanced S. miltiorrhiza growth in vitro but hindered biomass accumulation in situ. These observations suggested that the growth-promoting effects of endophytic fungi on S. miltiorrhiza were affected by the cultivation conditions changes.

Figure 1 The growth-promoting effect of endophytic fungi.

Effects of nine endophytic fungi on the net weight gain (A), increase in height (B), and number of new leaves (C) of S. miltiorrhiza seedlings in vitro culture. CK was a blank treatment without any endophytic fungi and cultured in the same way. The ability of nine endophytic fungi to secrete IAA, CK is a blank liquid medium (D). Effects of nine endophytic fungi on the dry weight of medicinal parts and biomass of S. miltiorrhiza after 36 weeks of pot culture (E). Vertical bars represent the SEM values, n = 3. An asterisk (*) indicates a statistical difference between the treatment and the control (* p < 0.05, ** p < 0.01).

Figure 2 The underground part of S. miltiorrhiza.

The underground part of S. miltiorrhiza after cocultivation with endophytic fungi for 36 weeks in the pot.

All nine endophytic fungi could secrete IAA, ranging from 0.68 to 2.29 mg L−1 (Fig. 1D). Additionally, no direct correlation between the S. miltiorrhiza growth index and the capacity of the endophytic fungi to produce IAA could be established regardless of in vitro or pot culture.

Beneficial endophytic fungi can promote photosynthesis in S. miltiorrhiza

After coculture in vitro 30 days, DS8 and 12 significantly enhanced the chlorophyll content. In contrast, the other six strains remarkably reduced the chlorophyll content, except DS7 (Fig. 3A). However, strains DS5 and 13 continued to exhibit marked growth-promoting effects, suggesting the involvement of pathway other than chlorophyll biosynthesis. Following 20 weeks of pot culture, DS5 and 16 markedly increased the chlorophyll content, in contrast to all the other strains except DS17 (Fig. 3B). Except DS13 and 16, all strains were able to boost the net photosynthetic rate (Pn) significantly (Fig. 3C). Moreover, all strains enhanced the transpiration rate (Tr) and stomatal conductance (Gs) in leaves (Figs. 3D and 3F). Likewise, the intercellular CO2 concentration (Ci) was markedly elevated by all strains except DS8 (Fig. 3E). A. sclerotigenum DS7 and 12 inhibited the synthesis of photosynthetic pigments, which may be one of the reasons for their detrimental effects on growth in S. miltiorrhiza.

Figure 3 Effects of nine strains of endophytic fungi on photosynthesis of Salvia miltiorrhiza.

Effect of endophytic fungi on the chlorophyll content of S. miltiorrhiza in vitro culture (A) and in situ culture (B). The effects of endophytic fungi on S. miltiorrhiza’s net photosynthetic rate (Pn) (C), transpiration rate (Tr) (D), intercellular carbon dioxide concentration (Ci) (E), and stomatal conductance (Gs) (F) after co-cultivation in the pot for 20 weeks. Vertical bars represent the SEM values, n = 3. An asterisk (*) indicates statistically significant differences between the treatment and the control (*p < 0.05, **p < 0.01).

Beneficial endophytic fungi can promote the C and N metabolism in S. miltiorrhiza

After coculture for 30 days, strains DS10, 12, and 13 conspicuously enhanced the activities of sucrose synthase (SS) and sucrose phosphate synthase (SPS) in the leaves. DS10 and 12 induced SPS activity more than SS, while DS13 did the opposite. DS7, 16, 17, and 18 significantly increased SS activity, while DS5 and 8 elevated that of SPS (Fig. 4A). All strains increased the activities of nitrate synthase (NR) and glutamine synthase (GS) in the leaves, especially markedly by DS5, 8, 10, 12, 13, 16, and 18 (Figs. 4C and 4D). The probiotic effects of DS5, 8, 10, 12, and 13 may be related to their promoting the synthesis and export of sucrose and N utilization. At the same time, DS7 only contributed to the promotion of sucrose synthesis and catabolism.

Figure 4 Effects of nine strains of endophytic fungi on C and N metabolism in Salvia miltiorrhiza.

Effect of nine endophytic fungi on the SS and SPS (A), NR (C), and GS (D) enzyme activities of S. miltiorrhiza after 30 days of in vitro culture. After 20 weeks of pot culture, the effects of nine endophytic fungi on SS and SPS (B), NR (F), GS (G), GOGAT (H) activity, soluble and reducing sugars (E), and soluble protein content (I) of S. miltiorrhiza leaves were examined. Vertical bars represent the SEM values, n = 3. An asterisk (*) indicates statistically significant differences between the treatment and the control (*p < 0.05, **p < 0.01).

After 20 weeks of potted culture, DS5, 7, 10, and 12 markedly enhanced SS activity; while DS7, 12, 13, 16, 17, and 18 that of SPS (Fig. 4B). Moreover, DS10 and 18 remarkably elevated the contents of only reducing sugars in leaves; while DS5, 7, 12, 13, 16, and 17 the contents of both soluble and reducing sugars (Fig. 4E). All nine endophytic fungi markedly increased the activity of NR; while DS8, 12, 13, and 18 significantly increased that of GS, unlike DS10 and 17 (Figs. 4F and 4G). DS7, 10, and 17 elevated GOGAT activity in leaves, especially DS10; in contrast to the other strains of which DS5 and 16 demonstrated the highest suppression (Fig. 4H). DS7, 12, and 13 promoted the accumulation of soluble proteins more notably in contrast to the other strains, unlike DS8 and 10 which inhibited it (Fig. 4I). DS7 promoted the accumulation by upregulating NR and GOGAT activities, while DS12 and 13 by upregulating NR and GS. DS8 may inhibit the accumulation by down-regulating GOGAT and DS10 the GS activity. The ability of DS8, 10, 12, 13, and 16 to elevate the NR and GS activities under both in vitro and in situ conditions suggested that the beneficial fungi assist in N assimilation.

In conclusion, DS5 and 10 demonstrated growth-promoting effects primarily by increasing sucrose catabolism and reducing the sugar content; DS13, 16, 17, and 18 through improving sucrose synthesis and sugar accumulation; and DS5, 8, 13, 16, and 18 through promoting N assimilation and the synthesis and accumulation of soluble proteins.

Beneficial endophytic fungi can promote resistance in S. miltiorrhiza

After 30 days of cocultivation, apart from DS16, all strains increased the SOD activity in S. miltiorrhiza leaves, with DS7, 10, and 18 exhibiting the most noticeable effects (Fig. 5A). All strains enhanced CAT activity, with DS12 exhibiting the most prominent increase; while DS13, 16, and 17 decreased it (Fig. 5B). While the other strains significantly reduced the POD activity, DS5, 7, 10, and 12 increased it, with DS7 and 10 showing the most notable impacts (Fig. 5C). Malondialdehyde (MDA) levels were reduced by all nine endophytic fungi, with DS5, 7, 8, 10, 12, and 13 inducing the most significant decline (Fig. 5E). DS5, 8, and 12 improved the PAL activity; whereas the other strains, particularly DS7, 13, and 18, significantly suppressed it (Fig. 5D). Interestingly, DS13 increased SOD activity while decreasing the activities of other antioxidant enzymes, but most significantly reduced MDA levels. These results suggested that DS13 improved the resistance of S. miltiorrhiza’s to stress primarily by enhancing the SOD activity and catalyzing the decomposition of O2−, which minimized the adverse effects of reactive oxygen species (ROS).

Figure 5 Effect of nine strains of endophytic fungi on the resistance system of Salvia miltiorrhiza cultured in vitro.

Effects of nine endophytic fungi on the antioxidant enzymes: SOD (A), CAT (B), POD (C), PAL (D), and MDA (E) contents of S. miltiorrhiza after 30 days of sterile culture. Vertical bars represent the SEM values, n = 3. An asterisk (*) indicates statistically significant differences between the treatment and the control (*p < 0.05, **p < 0.01).

Each of the nine strains significantly improved the SOD and PAL activities in the leaves of S. miltiorrhiza after 20 weeks of cocultivation (Figs. 6A and 6D). Except for strain DS16, which induced a decrease in CAT activity, all strains markedly improved it (Fig. 6B). Except for DS12 and 18, all the strains greatly boosted POD activity, while DS13 and 16 markedly decreased it (Fig. 6C). All strains except DS5, 7, 13, and 18 reduced the MDA content in leaves, while DS5 and 13 increased it (Fig. 6E). Except for DS13, 16, and 17, all strains were capable of elevating the Pro concentrations, with DS5, 7, 8, and 10 impacting the most noticeably (Fig. 6F). These results indicated that the strains DS8 and 10 can raise the activities of different antioxidant enzymes, lower the MDA levels, and increase the Pro content. Thus, these fungi benefit by scavenging free radicals, stabilizing the subcellular structures, maintaining normal cell physiology, metabolism, and morphology, and performing the ecologically important role of enhancing resistance.

Figure 6 Effects of nine strains of endophytic fungi on the resistance system of Salvia miltiorrhiza in situ culture.

Effect of endophytic fungi on the activities of the antioxidant enzymes SOD (A), CAT (B), POD (C), PAL (D), and the contents of MDA (E) and the osmoregulator Pro (F) in potted S. miltiorrhiza. Vertical bars represent the SEM values, n = 3. An asterisk (*) indicates statistically significant differences between the treatment and the control (*p < 0.05, **p < 0.01).

Beneficial endophytic fungi promote the synthesis of bioactive ingredients in the roots of S. miltiorrhiza

After 36 weeks of cocultivation in pots, the tanshinone I content was markedly enhanced by DS10 and 12 (Fig. 7A,); those of tanshinone IIA and cryptotanshinone by DS8, 10, and 12 (Figs. 7B, 7D); and those of dihydrotanshinone at 24.2- and 25.1-fold by DS5 and 7 respectively (Fig. 7C). All nine strains were able to increase the caffeic acid content in the roots (Fig. 7H), while DS16 and 5 were especially effective in enhancing the contents of rosmarinic acid and salvianolic acid B (Fig. 7F, and Fig. 7G). In summary, DS8, 10, and 12 were beneficial for the biosynthesis and accumulation tanshinone (apart from dihydrotanshinone) in the roots but not for phenolic acids (apart from caffeic acid). However, DS5 and 16 demonstrated the opposite effects. Additionally, A. sclerotigenum DS7 and 12 revealed the same facilitative or inhibitory effects on phenolic acids [apart from salvianolic acid A] (Figs. 7E, 7F, 7G and 7H), demonstrating the stable effects of these strains on the synthesis of phenolic acids and the regulation of growth.

Figure 7 Effects of nine strains of endophytic fungi on the content of bioactive ingredients in Salvia miltiorrhiza roots.

Effect of endophytic fungi on the content of fat-soluble and water-soluble bioactive ingredients in S. miltiorrhiza roots: tanshinone I (A), tanshinone IIA (B), dihydrotanshinone (C), cryptotanshinone (D), salvianolic acid A (E), salvianolic acid B (F), rosmarinic acid (G), and caffeic acid (H). Vertical bars represent the SEM values, n ≥ 10. An asterisk (*) indicates statistically significant differences between the treatment and the control (*p < 0.05, **p < 0.01).

The accumulation of bioactive ingredients in the roots of S. miltiorrhiza was calculated based on the average concentration of bioactive ingredients multiplied by the dry weight of the medicinally essential parts. Apart from DS5, which was unfavorable for the bioaccumulation of tanshinone I and cryptotanshinone, all strains were able to increase that of tanshinones (Fig. 8A). In comparison to the control group, DS12 enhanced tanshinone I bioaccumulation by up to 3.3-fold, while DS8 and 10 increased that of cryptotanshinone by up to 2.5–2.6-fold, respectively. DS5 and 7 conspicuously elevated the bioaccumulation of dihydrotanshinone, while DS16 and 18 that of tanshinones (Fig. 8A). All strains, except DS7, 8, and 10, enhanced the bioaccumulation of salvianolic acid A (Fig. 8B). DS16 that of phenolic acids; and DS7, 8, and 12 that of tanshinones but inhibited that of rosmarinic acid and salvianolic acid B. DS5 inhibited the accumulation of three tanshinones but promoted that of phenolic acid (Figs. 8A and 8B).

Figure 8 Effect of nine strains of endophytic fungi on the accumulation of bioactive ingredients in the roots of S. miltiorrhiza.

Effects of nine endophytic fungi strains on the accumulation of fat-soluble (A) and water-soluble (B) bioactive ingredients in the roots of S. miltiorrhiza.

Discussion

The growth-promoting effect of endophytic fungi

Endophytic fungi enhanced the growth and development of various crops and medicinal plants such as wheat and tomatoes (Afridi et al., 2019; Nefzi et al., 2019), S. miltiorrhiza (Vandana et al., 2021), Dendrobium officinale (Hassan, 2017; Hou & Guo, 2014), Centella asiatica (Satheesan, Narayanan & Sakunthala, 2012), and Ammopiptanthus mongolicus (Li et al., 2018a). The inoculation of S. miltiorrhiza with endophytic fungi such as Paecilomyces sp. (Tang, Li & Guo, 2014) and Cladosporium sp. SM58 (Zhou, Tang & Guo, 2018a) improved growth and yield. In this study, four strains of endophytic fungi displayed growth-promoting effects in vitro, while seven showed similar results in situ. However, it is pertinent to note that the impacts of endophytic fungi observed under controlled lab conditions may not necessarily be applied to field conditions (De Lamo & Takken, 2020). For instance, the growth-promoting effects of inoculating endophytic fungi to Miscanthus were not observed in pot experiments but were significant in field experiments (Schmidt et al., 2018). In this research, Acremonium sclerotigenum DS7 and 12 demonstrated practical growth-promoting effects under sterile conditions but not in pot tests, suggesting that the cultivation conditions influenced the regulatory effects of endophytic fungi on plant growth. DS16 demonstrated significant growth-promoting effects and yield-enhancing potential on S. miltiorrhiza in pots, but additional field experiments are necessary for confirmation. The ability of microorganisms to secrete IAA (Indole-3-acetic acid) is frequently used as an indicator for screening beneficial fungi (Lobo et al., 2022). However, this investigation demonstrated that the growth-promoting ability of endophytic fungi is not necessarily related to IAA secretion, consistent with the results of research on tomatoes and ephedra (Cochard et al., 2022; Khalil et al., 2021). Thus, it is suggested that the selection of beneficial microorganisms should not rely solely on their capacity to release IAA.

Endophytic fungi and C and N assimilation in plants

Endophytic fungi significantly impact the growth of host plants through various mechanisms, such as enhancing the photosynthetic capacity, promoting C and N assimilation, and secreting phytohormones (Wani et al., 2015). The study found that strains DS5 and 16 improved the chlorophyll content of S. miltiorrhiza plants grown in pots, while strains DS7, 8, and 12 under sterile conditions. A high chlorophyll concentration has been linked with enhanced yield (Chen et al., 2018). Similarly, the current study showed that the group of Salvia plants with increased chlorophyll content had a higher biomass. In addition, under drought, inoculation with endophytic fungus increased the Pn and Ci levels in the leaves (Martínez-Arias et al., 2021) and soluble sugars in the roots (Sun et al., 2022) of host plants. All nine strains in this study effectively increased stomatal conductance and leaf transpiration rate, while most strains considerably enhanced the net photosynthetic rate and intercellular CO2 concentration. All strains, except DS8, promoted the accumulation of soluble and reducing sugars. In summary, endophytic fungi enhanced the production of photosynthetic pigments in the host plant and improved photosynthetic efficiency and C assimilation, ultimately promoting the buildup of organic matter.

Sucrose metabolism, catalyzed by the key enzymes SS and SPS; and N assimilation, catalyzed by the major enzymes NR, GS, and GOGAT, are pivotal for plant growth. Moreover, coordinating photosynthetic C assimilation and N uptake and utilization is critical to ensuring plant yield and quality (Illescas et al., 2022). This study found that under sterile conditions, DS7, 8, 10, 12, and 13 enhanced SS, SPS, NR, and GS activities and showed growth-promoting effects. In potted cultivation, DS5, 8, 13, 16, and 18 elevated SS, SPS, NR, and GS activities but inhibited GOGAT activity while exerting growth-promoting effects; DS10 and 17 elevated the SS, SPS, NR, and GOGAT activities, but inhibited GS activity while exerting growth-promoting effects. DS7, 12, and 13 significantly enhanced the contents of soluble proteins, soluble sugars, and reducing sugars under both in vitro and in situ conditions, but only strain DS13 exhibited a growth-promoting impact in pots. Although the production of assimilates by DS7 and 12 could have been used in other metabolic pathways without exhibiting any growth-promoting effects, the strains enhanced the absorption and assimilation of C and N by the host plant.

Different endophytic fungi regulated C and N metabolism in host plants with various characteristics. They can enhance the biomass of cocoa trees by promoting the uptake and distribution of N (Christian, Herre & Clay, 2019). Inoculating Codonopsis pilosula with Trichoderma led to an increased SS activity (Wang et al., 2021) and Achnatherum inebrians with Epichloë gansuensis that of increased the GS and NiR (Wang et al., 2018), ultimately benefiting biomass. In this study, DS10 and 13 enhanced intercellular CO2 concentration; net photosynthetic rate; activities of SS, SPS, NR, and GS; and improved the efficiency of C and N assimilation, consequently promoting S. miltiorrhiza growth. Additionally, DS16 elevated the production of photosynthetic pigments; raised intercellular CO2 concentrations; and activated SPS, NR, and GS enzymes, resulting in significant improvements in C and N assimilation efficiency and a coordinated distribution of C and N metabolites within the plant. Endophytic fungi primarily promote plant growth by improving photosynthesis, activating C and N assimilation, balancing C and N metabolism, and increasing the demand of plants for energy resources. However, the mechanisms by which various beneficial endophytic fungi exert their growth-promoting effects vary. Some endophytic fungi may be involved only in a fraction of plant metabolism and, therefore, may not have any noticeable growth-promoting effects (Fang, Fernie & Luo, 2019). Therefore, a potential strategy to maximize the growth-promoting characteristics of beneficial endophytic fungi must involve constructing artificial communities by selecting strains that perform different ecological functions.

Endophytic fungi and plant resistance

During the growth and development of plants, various biotic and abiotic factors in the environment can lead to the intra-cellular production of large amounts of ROS. High concentrations of ROS can cause lipid peroxidation, membrane degradation, and damage the proteins, lipids, and nucleic acids in plants (Fan, Subramanian & Smith, 2020). The plant antioxidant system, through the combined action of antioxidant enzymes such as superoxide dismutase (SOD), peroxidase (POD), catalase (CAT), and non-enzymatic antioxidants such as malondialdehyde (MDA), proline (Pro), and soluble proteins, eliminates ROS and maintains normal growth and development (Zhu et al., 2022).

The ROS scavenging system established by inoculating endophytic fungi enables plants to withstand major abiotic stresses such as salinity, drought, high temperature, and heavy metals and also protects them from pathogen attacks (Gill et al., 2016). Piriformospora indica inoculated into Citrus sinensis activated the antioxidant defense system of the host plants, improving their resistance during field cultivation (Li et al., 2022). In this study, the inoculation of nine endophytic fungi significantly increased the activities of SOD and PAL in S. miltiorrhiza, consistent with those in wheat leaves after inoculation with Trichoderma asperellum, effectively enhancing drought resistance (Illescas et al., 2022). DS8, 10, and 16 significantly improved the activities of SOD, PAL, CAT, and POD, and Pro contents; while reducing MDA contents under pot cultures. This was consistent with the results in Codonopsis pilosula after inoculation with Trichoderma strain RHTA01, ultimately alleviating the damage caused by abiotic stress (Wang et al., 2021). In addition, endophytic fungi can induce systemic resistance (SR) in host plants by triggering the transmission of signaling molecules such as jasmonic acid (JA) and ethylene (ET) (De Lamo & Takken, 2020). SR induced by endophytic fungi was significant in enhancing plant resistance to environmental stress and pathogen invasion, as well as promoting plant growth and yield (De la Vega-Camarillo et al., 2023). The endophytic fungus Piriformospora indica induces plant stress tolerance by enhancing the activity of antioxidant enzymes and the expression levels of defense- and stress-related genes(Khalid, Rahman & Huang, 2019). In conclusion, when endophytic fungi colonize plants, they induce the antioxidant system which is beneficial for enhancing the adaptability and resistance to the environmental stresses, which is of great significance for the healthy growth of S. miltiorrhiza.

Endophytic fungi and plant bioactive ingredients

Endophytic fungi can promote the synthesis and accumulation of secondary metabolites in host plants (Zhai et al., 2017a). This study found that DS5, 7, 8, 10, and 12 could elevate the contents of tanshinones in the roots of S. miltiorrhiza, of which DS8, 10, and 12 demonstrated the most significant effects. Similar results were observed with Mucor circinelloides DF20 (Chen et al., 2021) and Chaetomium globosum D38 (Zhai et al., 2017b), despite differences in their origin. M. circinelloides DF20 upregulated enzymes like DXS and DXR, critical for tanshinone biosynthesis and accumulation in roots. Moreover, polysaccharides produced by endophytic fungi stimulated tanshinone production and the growth of plants (Ming et al., 2013). Thus, the beneficial endophytic fungi not only regulated necessary enzymes but also promoted the growth and the synthesis of plant secondary metabolites in general.

DS5 and 16 increased the bioaccumulation of the phenolic acids salvianolic acid B and rosmarinic acid, akin to the effects of Alternaria sp. A13 in S. miltiorrhiza roots (Zhou, Tang & Guo, 2018b). The activity of PAL positively correlated with phenolic acid content (Jaafar, Ibrahim & Fakri, 2012); DS5 and 16 upregulated PAL activity in the roots, leading to the enhanced accumulation of phenolic acids. Rosmarinic acid is a crucial precursor for the synthesis of salvianolic acid (Shi et al., 2019); the promotion of rosmarinic acid biosynthesis by DS5 and 16 can further stimulate the synthesis of salvianolic acid. DS5 considerably increased the content and bioaccumulation of dihydrotanshinone, which has pharmacological properties such as antitumor, cardiovascular protection, and bacterial inhibition; it may have application potential in the pharmaceutical industry.

Plant defense is an essential process that requires both biomaterials and energy, with secondary metabolites playing a vital role in defense (Caretto et al., 2015). However, due to limited resources, a trade-off between growth and defense is persistent throughout all growth and development stages in plants (He, Webster & He, 2022). Endophytic fungi regulate the secondary metabolism in plants in various ways. In this experiment, DS8, 10, and 12 enhanced the tanshinone biosynthesis and accumulation but not phenolic acid (except caffeic acid). On the contrary, DS5 significantly promoted phenolic acid component biosynthesis and accumulation but not tanshinones (except dihydrotanshinone). In this experiment, no single endophytic fungus could elevate the content of all bioactive ingredients, similar to the changes in phenolics and volatiles after inoculating three endophytic fungi in Houttuynia cordata Thunb (Ye et al., 2021). Additionally, in this experiment, when a particular strain increased the contents of the bioactive ingredients, it did not necessarily enhance the bioaccumulation. Promoting the biosynthesis of secondary metabolites increased the production of biomass and energy consumption, which affected biomass accumulation (Aharoni & Galili, 2011). Therefore, to achieve the desired results, the trade-off between yield and the production of the target bioactive ingredients should be considered when cocultured with endophytic fungi. Therefore, a pre-designed microbial community approach based on the different characteristics of the strains may be an effective means of increasing the content and bioaccumulation of the target bioactive ingredients in medicinal plants.

Conclusions

Endophytic fungi are a crucial biological resource that enhanced the yield and quality of the medicinal herb S. miltiorrhiza ’s and improved its environmental adaptability. The different genera, species, or strains of these fungi regulated the growth, development, and metabolism of S. miltiorrhiza in unique ways. For instance, DS5, 7, 8, 10, 12, and 16 demonstrated various beneficial impacts on S. miltiorrhiza, such as promoting growth, increasing resistance, and boosting the tanshinone or salvianolic acid contents. Although this study did not identify a single strain that could enhance both the yield and the contents of tanshinone and salvianolic acids in S. miltiorrhiza, the strains used were vital for utilizing the artificially modulated microbial communities to balance the relationship between yield, quality, and productivity in salvia. This investigation also suggested that the most effective way to maximize the ecological functions of beneficial microorganisms and improve the productivity of medicinal plants for selecting specific strains of pre-designed microbial communities based on the different ecological functions of endophytic fungi under various environments and distinct production goals.

Supplemental Information

Figure S1 Colonization of nine strains of endophytic fungi in Salvia miltiorrhiza

Figure S2 UPLC chromatogram of mixed standards (a) and sample (b)

3: Caffeic acid; 4: Rosmarinic acid; 5: Salvianolic acid B; 6: Salvianolic acid A; 7: Dihydrotanshinone; 8: Tanshinone I; 9: Cryptotanshinone; 10: Tanshinone IIA.

Table S1 The results of the system suitability evaluation

Supplemental Information 4 Standard curves for different bioactive ingredients

Y-axis represents peak area and X-axis represents sample size. Figure headings represent the different active ingredients.

Supplemental Information 5 Sequence information of nine strains of endophytic fungi

Supplemental Information 6 Raw data

Supplemental Information 7 Supplementary data

Photographs of in vitro and greenhouse cultures and photographs of Salvia miltiorrhiza roots in the experiment

We thank Bullet Edits Limited for the linguistic editing and proofreading of the manuscript. The special acknowledgments are given to the editors and to the reviewers.

Additional Information and Declarations

Competing Interests

Author Contributions

DNA Deposition

Data Availability

The authors declare there are no competing interests. Yali Lin is employed by Patent Examination Cooperation Sichuan Center of the Patent Office.

Xiaoyu Li conceived and designed the experiments, analyzed the data, prepared figures and/or tables, authored or reviewed drafts of the article, and approved the final draft.

Yali Lin conceived and designed the experiments, performed the experiments, analyzed the data, prepared figures and/or tables, and approved the final draft.

Yong Qin analyzed the data, prepared figures and/or tables, and approved the final draft.

Guiqi Han conceived and designed the experiments, authored or reviewed drafts of the article, and approved the final draft.

Hai Wang conceived and designed the experiments, authored or reviewed drafts of the article, and approved the final draft.

Zhuyun Yan conceived and designed the experiments, authored or reviewed drafts of the article, and approved the final draft.

The following information was supplied regarding the deposition of DNA sequences:

The ITS sequences of the nine endophytic fungal strains in this experiment are available at GenBank: OP547504–OP547512.

The following information was supplied regarding data availability:

The raw data are available in the Supplemental File.

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
