# Peer review of "Beneficial endophytic fungi improve the yield and quality of Salvia miltiorrhiza by performing different ecological functions"

_PeerJ, doi:10.7717/peerj.16959_

## Round 0.1 · original submission · Major Revisions

Thank you for your submission to PeerJ. Please see the helpful suggestions and comments by the reviewers in revising your manuscript. In particular, both reviewers noted that additional details regarding methodology are needed to support your conclusions. I look forward to receiving the revised manuscript.

**Language Note:** The review process has identified that the English language must be improved. PeerJ can provide language editing services - please contact us at copyediting@peerj.com for pricing (be sure to provide your manuscript number and title). Alternatively, you should make your own arrangements to improve the language quality and provide details in your response letter. – PeerJ Staff

·

Basic reporting

There were minor grammatical errors throughout the manuscript. For the most part, ideas were communicated clearly, but having a fluent English speaker read over the manuscript and correct these grammatical errors would improve the clarity. Paragraph two of the introduction (lines 84-113) was lacking in citations. Specifically, lines 84-96 contained no in-text citations, but I did not feel the information included was “common knowledge”.

Experimental design

Overall, the experimental design was well described. However, in the first paragraph of the section “Experimental design” (lines 154-165), the use of control plants was not described. The number of control plants, if they were punctured with the sterile needle, etc. was not clearly included here but control plants were referenced later in the statistical reporting and the figures. Additionally, it was stated in the next paragraph that control plants were used in the “Greenhouse potted culture” experiment, but additional details on how these plants were treated could have been included. Lastly, the section “Determination of active ingredients of S. miltiorrhiza” (lines 302-306), was lacking in detail. It was clear that the active ingredients of S. miltiorrhiza was identified using liquid chromatography, but beyond that was unclear. In addition to the citation indicating where you found methods, the specific method of identifying the active ingredient should be included (i.e. was it based only on retention time, was there another colorimetric assay, etc.).

Validity of the findings

No comment

Reviewer 2 ·

Basic reporting

Background of this manuscript is described straight-forward and sufficient. Authors obtained an extensive data set to understand the effect of endophytes. There are some parts where English is not clear and professional (please see line-by-line comments). Please provide citations with the lines 77, 86, and 99. Authors supplied raw data and supplementary figures.

Experimental design

Line 310: Please provide explanations why LSD (Fischer’s?) is preferred to other post-hoc multiple comparison methods such as Dunnett. LSD has an issue of controlling type I error in multiple comparison (Ruxton and Beauchamp, Behav. Ecol., 2008).

Fig.1: Please add some explanation of CK in the legend.

Fig.8: I suggest arranging y-axis in the same order as other figures.

Validity of the findings

Line 319 and 325: I do not understand this part. Fig. 1e showed only DS16 strain promoted dry weight and biomass. Please provide other data to support that DS8 promoted plant growth.

Line 37, and 385 - 414: I suggest revising your discussion in this chapter to avoid extrapolation. Can the observation in this study result from the resistance of host plants to these endophytes but not related to plant resistance enhancement? To support the claim that these endophytes enhance disease resistance, please provide data, such as a decrease in disease incidence after inoculation with pathogen, or discuss other possibilities.

Line 456-468: I agree that environmental factors can affect the symbiosis between plants and endophytes in general. However, I am not convinced by the data presented in this manuscript because the inoculation/cultivation conditions of the in-vitro and pot assays are different, and there was no direct comparison to test the effect of environmental factors such as temperature. I suggest specifying this part using the words such as the effect of inoculation/cultivation condition.

Additional comments

The objective of this study is clear, and the findings are interesting for achieving increased production of active compounds. Nevertheless, several parts of the conclusions mentioned in <Validity of findings> section of this review are vague and not convincing due to the lack of experiments to test your hypothesis. From this perspective, I suggest narrowing down discussions such as cultivation conditions.

<Line-by-line comment>

Line 57: Definition of endophytic fungi is vague and too broad.

Line 78-80: What does “to enhance active ingredient” mean?

Line 96: Salvia -> S.

Line 104: T. -> Trichoderma?

Line 154: Culture -> culture

Line 163-165: Pictures or any index of fungal colonization would be helpful for readers to compare the manner of infection and colonization rate of each endophyte.

Line 513: Epichloë

Line 585: Please clarify "plant matter"

Fig 1 b: hight -> height

References: Italicize species name in the titles

Line 629, 700: Journal name

Line 726: Inappropriate line break

---

## Round 0.2 · Minor Revisions

Thank you for your revised manuscript. Both reviewers commented that the manuscript is much improved; however, there are a few remaining issues raised by Reviewer 2. Please see their suggestions, including the line-by-line suggestions. I look forward to receiving your revised submission.

·

Basic reporting

The language was much clearer and easier to understand. I appreciate the addition of references and found them to be sufficient this time around.

Experimental design

The method section was much clearer on a second reading and the addition of citations will be helpful for future work wanting to achieve similar results.

Validity of the findings

I found the the Results and the Discussion sections to be much more robust and linked to additional supporting research. Nice work.

Additional comments

Thank you for taking the time to consider the comments and make appropriate edits.

Reviewer 2 ·

Basic reporting

The authors added citations and corrected most of the grammatical errors.
Please see line-by-line comments for minor errors in the Additional comments section.

Experimental design

The authors addressed my concern in the revised manuscript.

Validity of the findings

The authors revised their discussion for clarity and addressed my concerns.

Additional comments

Figure S1: The added figure is helpful. Thank you. Adding arrows or triangles to indicate endophytes in this figure would be great for clarification.

Line 171-173: Thank you for your revision. The authors discuss only vertically-transmitted endophytes, but I suppose the strains that were used in the study are not necessarily vertically-transmitted.
Also, readers would be confused about the sentence “the spread of spores and mycelial fragments between different plants, thereby colonizing a wide range of taxa.” Please revise for clarity.

Line 508: Please italicize Citrus sinensis.

Line 520-522: The authors focused on endophytic fungi in this discussion so I suggest revising the sentence not to emphasize bacterial species but to focus on the importance of antioxidant enzymes on disease resistance to avoid confusion.

Figure 9: This figure should be included as Supplementary Files/Figures with more details of mixed standards and samples. It does not directly support the conclusion.

---

## Round 0.3 · accepted · Accept

Thank you for revising the manuscript with these last suggested changes. I believe your manuscript will be of interest to many people and it is a nice contribution to the literature in this area.